# CAN I TRUST THE EXPLAINER?
# VERIFYING POST-HOC EXPLANATORY METHODS

## ABSTRACT

For AI systems to garner widespread public acceptance, we must develop methods capable of explaining the decisions of black-box models such as neural networks. In this work, we identify two issues of current explanatory methods. First, we show that two prevalent perspectives on explanations—*feature-additivity* and *feature-selection*—lead to fundamentally different instance-wise explanations. In the literature, explainers from different perspectives are currently being directly compared, despite their distinct explanation goals. The second issue is that current post-hoc explainers are either validated under simplistic scenarios (on simple models such as linear regression, or on models trained on syntactic datasets), or, when applied to real-world neural networks, explainers are commonly validated under the assumption that the learned models behave reasonably. However, neural networks often rely on unreasonable correlations, even when producing correct decisions. We introduce a verification framework for explanatory methods under the feature-selection perspective. Our framework is based on a non-trivial neural network architecture trained on a real-world task, and for which we are able to provide guarantees on its inner workings. We validate the efficacy of our evaluation by showing the failure modes of current explainers. We aim for this framework to provide a publicly available,[1] off-the-shelf evaluation when the feature-selection perspective on explanations is needed.

## 1 INTRODUCTION

A large number of post-hoc explanatory methods have recently been developed with the goal of shedding light on highly accurate, yet black-box machine learning models (Ribeiro et al., 2016a; Lundberg & Lee, 2017; Arras et al., 2017; Shrikumar et al., 2017; Ribeiro et al., 2016b; 2018; Plumb et al., 2018; Chen et al., 2018). Among these methods, there are currently at least two widely used perspectives on explanations: *feature-additivity* (Ribeiro et al., 2016a; Lundberg & Lee, 2017; Shrikumar et al., 2017; Arras et al., 2017) and *feature-selection* (Chen et al., 2018; Ribeiro et al., 2018; Carter et al., 2018), which we describe in detail in the sections below. While both shed light on the overall behavior of a model, we show that, when it comes to explaining the prediction on a single input in isolation, i.e., *instance-wise explanations*, the two perspectives lead to fundamentally different explanations. In practice, explanatory methods adhering to different perspectives are being directly compared. For example, Chen et al. (2018) and Yoon et al. (2019) compare L2X, a feature-selection explainer, with LIME (Ribeiro et al., 2016a) and SHAP (Lundberg & Lee, 2017), two feature-additivity explainers. We draw attention to the fact that these comparisons may not be coherent, given the fundamentally different explanation targets, and we discuss the strengths and limitations of the two perspectives.

Secondly, while current explanatory methods are successful in pointing out catastrophic biases, such as relying on headers to discriminate between pieces of text about Christianity and atheism (Ribeiro et al., 2016a), it is an open question to what extent they are reliable when the model that they aim to explain (which we call the *target model*) has a less dramatic bias. This is a difficult task, precisely because the ground-truth decision-making process of neural networks is not known. Consequently, when applied to complex neural networks trained on real-world datasets, a prevalent way to evaluate the explainers is to assume that the target models behave reasonably, i.e., that they did not rely

---

[1]Code, generated datasets, and trained models will be released.

on irrelevant correlations. For example, in their morphosyntactic agreement paradigm, Pörner et al. (2018) assume that a model that predicts if a verb should be singular or plural given the tokens before the verb, must be doing so by focusing on a noun that the model had identified as the subject. Such assumptions may be poor, since recent works show a series of surprising spurious correlations in human-annotated datasets, on which neural networks learn to heavily rely (Gururangan et al., 2018; Glockner et al., 2018; Carmona et al., 2018). Therefore, it is not reliable to penalize an explainer for pointing to tokens that just do not appear significant to us.

We address the above issue by proposing a framework capable of generating evaluation tests for the explanatory methods under the feature-selection perspective. Our tests consist of pairs of (target model, dataset). Given a pair, for each instance in the dataset, the specific architecture of our model allows us to identify a subset of tokens that have zero contribution to the model's prediction on the instance. We further identify a subset of tokens clearly relevant to the prediction. Hence, we test if explainers rank zero-contribution tokens higher than relevant tokens. We instantiated our framework on three pairs of (target model, dataset) on the task of multi-aspect sentiment analysis. Each pair corresponds to an aspect and the three models (of same architecture) have been trained independently. We highlight that our test is not a sufficient test for concluding the power of explainers in full generality, since we do not know the whole ground-truth behaviour of the target models. Indeed, we do not introduce an explanation generation framework but a framework for generating evaluation tests for which we provide certain guarantees on the behaviour of the target model. Under these guarantees we are able to test the explainers for critical failures. Our framework therefore generates necessary evaluation tests, and our metrics penalize explainers only when we are able to guarantee that they produced an error. To our knowledge, we are the first to introduce an automatic and non-trivial evaluation test that does not rely on speculations on the behavior of the target model.

Finally, we evaluate L2X (Chen et al., 2018), a feature-selection explainer, under our test. Even though our test is specifically designed for feature-selection explanatory methods, since, in practice, the two types of explainers are being compared, and, since LIME (Ribeiro et al., 2016a) and SHAP (Lundberg & Lee, 2017) are two very popular explainers, we were interested in how the latter perform on our test, even though they adhere to the feature-additivity perspective. Interestingly, we find that, most of the time, LIME and SHAP perform better than L2X. We will detail in Section 5 the reasons why we believe this is the case. We provide the error rates of these explanatory methods to raise awareness of their possible modes of failure under the feature-selection perspective of explanations. For example, our findings show that, in certain cases, the explainers predict the most relevant token to be among the tokens with zero contribution. We will release our test, which can be used off-the-shelf, and encourage the community to use it for testing future work on explanatory methods under the feature-selection perspective. We also note that our methodology for creating this evaluation is generic and can be instantiated on other tasks or areas of research.

## 2 RELATED WORK

The most common instance-wise explanatory methods are *feature-based*, i.e., they explain a prediction in terms of the input unit-features (e.g., tokens for text and super-pixels for images). Among the feature-based explainers, there are two major types of explanations: (i) feature-additive: provide signed weights for each input feature, proportional to the contributions of the features to the model's prediction (Ribeiro et al., 2016a; Lundberg & Lee, 2017; Shrikumar et al., 2017; Arras et al., 2017), and (ii) feature-selective: provide a (potentially ranked) subset of features responsible for the prediction (Chen et al., 2018; Ribeiro et al., 2018; Carter et al., 2018). We discuss these explanatory methods in more detail in Section 3. Other types of explanations are (iii) example-based (Koh & Liang, 2017): identify the most relevant instances in the training set that influenced the model's prediction on the current input, and (iv) human-level explanations (Camburu et al., 2018; Park et al., 2018; Kim et al., 2018; Bekele et al., 2018): explanations that are similar to what humans provide in real-world, both in terms of arguments (human-biases) and form (full-sentence natural language). In this work, we focus on verifying feature-based explainers, since they represent the majority of current works.

While many explainers have been proposed, it is still an open question how to thoroughly validate their faithfulness to the target model. There are four types of evaluations commonly performed:

1. **Interpretable target models.** Typically, explainers are tested on linear regression and decision trees (e.g., LIME (Ribeiro et al., 2016a)) or support vector representations (e.g., MAPLE (Plumb et al., 2018)). While this evaluation accurately assesses the faithfulness of the explainer to the target model, these very simple models may not be representative for the large and intricate neural networks used in practice.

2. **Synthetic setups.** Another popular evaluation setup is to create synthetic tasks where the set of important features is controlled. For example, L2X (Chen et al., 2018) was evaluated on four synthetic tasks: 2-dim XOR, orange skin, nonlinear additive model, and switch feature. While there is no limit on the complexity of the target models trained on these setups, their synthetic nature may still prompt the target models to learn simpler functions than the ones needed for real-world applications. This, in turn, may ease the job for the explainers.

3. **Assuming a reasonable behavior.** In this setup, one identifies certain intuitive heuristics that a high-performing target model is assumed to follow. For example, in sentiment analysis, the model is supposed to rely on adjectives and adverbs in agreement with the predicted sentiment. Crowd-sourcing evaluation is often performed to assert if the features produced by the explainer are in agreement with the model's prediction (Lundberg & Lee, 2017; Chen et al., 2018). However, neural networks may discover surprising artifacts (Gururangan et al., 2018) to rely on, even when they obtain a high accuracy. Hence, this evaluation is not reliable for assessing the faithfulness of the explainer to the target model.

4. **Are explanations helping humans to predict the model's behaviour?** In this evaluation, humans are presented with a series of predictions of a model and explanations from different explainers, and are asked to infer the predictions (outputs) that the model will make on a separate set of examples. One concludes that an explainer $E1$ is better than an explainer $E2$ if humans are consistently better at predicting the output of the model after seeing explanations from $E1$ than after seeing explanations from $E2$ (Ribeiro et al., 2018). While this framework is a good proxy for evaluating the real-world usage of explanations, it is expensive and requires considerable human effort if it is to be applied on complex real-world neural network models.

In contrast to the above, our evaluation is fully automatic, the target model is a non-trivial neural network trained on a real-world task and for which we provide guarantees on its inner-workings. Our framework is similar in scope with the sanity check introduced by Adebayo et al. (2018). However, their test filters for the basic requirement that an explainer should provide different explanations for a model trained on real data than when the data and/or model are randomized. Our test is therefore more challenging and requires a stronger fidelity of the explainer to the target model.

## 3 INSTANCE-WISE EXPLANATIONS

As mentioned before, current explanatory methods adhere to two major perspectives of explanations:

***Perspective 1 (Feature-additivity)***: For a model $f$ and an instance $x$, the explanation of the prediction $f(x)$ consists of a set of contributions $\{w_i^x(f)\}_i$ for each feature $i$ of $x$ such that the sum of the contributions of the features in $x$ approximates $f(x)$, i.e., $\sum_i w_i^x(f) \approx f(x)$.

Many explanatory methods adhere to this perspective (Arras et al., 2017; Shrikumar et al., 2017; Ribeiro et al., 2016a). For example, LIME (Ribeiro et al., 2016a) learns the weights via a linear regression on the neighborhood (explained below) of the instance. Lundberg & Lee (2017) unified this class of methods by showing that the only set of feature-additive contributions that verify three desired constraints (local accuracy, missingness, and consistency—we refer to their paper for details) are given by the Shapley values from game theory:

$$w_i^x(f) = \sum_{x' \in x \setminus \{i\}} \frac{|x'|!(|x| - |x'| - 1)!}{|x|!} [f(x' \cup \{i\}) - f(x')], \qquad (1)$$

where the sum enumerates over all subsets $x'$ of features in $x$ that do not include the feature $i$, and $|\cdot|$ denotes the number of features of its argument.

Thus, the contribution of each feature $i$ in the instance $x$ is an *average* of its contributions over a neighborhood of the instance. Usually, this neighborhood consists of all the perturbations given by

M: IF *"very good"* IN *input*: RETURN $0.9$;
  IF *"nice"* IN *input*: RETURN $0.7$;
  IF *"good"* IN *input*: RETURN $0.6$;
  RETURN $0$.

$x_1$: "The movie was good, it was actually nice."
$M(x_1) = 0.7$

| *Feature-additivity* | *Feature-selection* |
|---|---|
| nice: 0.4 | {*nice*} |
| good: 0.3 | |
| rest of tokens: 0 | |

$x_2$: "The movie was nice, in fact, it was very good."
$M(x_2) = 0.9$

| *Feature-additivity* | *Feature-selection* |
|---|---|
| good: 0.417 | {*good*, *very*} |
| nice: 0.367 | |
| very: 0.116 | |
| rest of tokens: 0 | |

Figure 1: Examples on which the two perspectives give different instance-wise explanations.

masking out combinations of features in $x$; see, e.g., (Ribeiro et al., 2016a; Lundberg & Lee, 2017). However, Laugel et al. (2018) show that the choice of the neighborhood is critical, and it is an open question what neighborhood is best to use in practice.

***Perspective 2 (Feature-selection)***: For a model $f$ and an instance $x$, the explanation of $f(x)$ consists of a sufficient (ideally small) subset $S(x)$ of (potentially ranked) features that alone lead to (almost) the same prediction as the original one, i.e., $f(S(x)) \approx f(x)$.

Chen et al. (2018); Carter et al. (2018), and Ribeiro et al. (2018) adhere to this perspective. For example, L2X (Chen et al., 2018) learns $S(x)$ by maximizing the mutual information between $S(x)$ and the prediction. However, it assumes that the number of important features per instance, i.e., $|S(x)|$, is known, which is usually not the case in practice. A downside of this perspective is that it may not always be true that the model relied only on a (small) subset of features, as opposed to using all the features. However, this can be the case for certain tasks, such as sentiment analysis.

To better understand the differences between the two perspectives, in Figure 1, we provide the instance-wise explanations that each perspective aims to provide for a hypothetical sentiment analysis regression model, where 0 is the most negative and 1 the most positive score. We note that our hypothetical model is not far, in behaviour, from what real-world neural networks learn, especially given the notorious biases in the datasets. For example, Gururangan et al. (2018) show that natural language inference neural networks trained on SNLI (Bowman et al., 2015) may heavily rely on the presence of a few specific tokens in the input, which should not even be, in general, indicators for the correct target class, e.g., "outdoors" for the entailment class, "tall" for the neutral class, and "sleeping" for the contradiction class.

In our examples in Figure 1, we clearly see the differences between the two perspectives. For the instance $x_1$, the feature-additive explanation tells us that "nice" was the most relevant feature, with a weight of $0.4$, but also that "good" had a significant contribution of $0.3$. While for this instance alone, our model relied only on "nice" to provide a positive score of $0.7$, it is also true that, if "nice" was not present, the model would have relied on "good" to provide a score of $0.6$. Thus, we see that the feature-additive perspective aims to provide an *average explanation* of the model on a *neighborhood* of the instance, while the feature-selective perspective aims to tell us the pointwise features used by the model on the instance *in isolation*, such as "nice" for instance $x_1$.

An even more pronounced difference between the two perspectives is visible on instance $x_2$, where the ranking of features is now different. The feature-selective explanation ranks "good" and "nice" as the two most important features, while on the instance $x_2$ in isolation, the model relied on the tokens "very" and "good", that the feature-selection perspective would aim to provide.

Therefore, we see that, while both perspectives of explanations give insights into the model's behavior, one perspective might be preferred over the other in different real-world use-cases. In the rest of the paper, we propose a verification framework for the *feature-selection* perspective of instance-wise explanations.

## 4 OUR VERIFICATION FRAMEWORK

Our proposed verification framework leverages the architecture of the RCNN model introduced by Lei et al. (2016). We further prune the original dataset on which the RCNN had been trained to ensure that, for each datapoint $x$, there exists a set of tokens that have zero contribution (irrelevant features) and a set of tokens that have a significant contribution (clearly relevant features) to RCNN's prediction on $x$. We further introduce a set of metrics that measure how explainers fail to rank the irrelevant tokens lower than the clearly revelant ones. We describe each of these steps in detail below.

**The RCNN.** The RCNN (Lei et al., 2016) consists of two modules: a generator followed by an encoder, both instantiated with recurrent convolutional neural networks (Lei et al., 2015). The generator is a bidirectional network that takes as input a piece of text $x$ and, for each of its tokens, outputs the parameter of a Bernoulli distribution. According to this distribution, the RCNN selects a subset of tokens from $x$, called $\mathcal{S}_x = \text{generator}(x)$, and passes it to the encoder, which makes the final prediction solely as a function of $\mathcal{S}_x$. Thus:

$$\text{RCNN}(x) = \text{encoder}(\text{generator}(x)) = \text{encoder}(\mathcal{S}_x). \tag{2}$$

There is no direct supervision on the subset selection, and the generator and encoder were trained jointly, with supervision only on the final prediction. The authors also used two regularizers: one to encourage the generator to select a short sub-phrase, rather than disconnected tokens, and a second to encourage the selection of fewer tokens. At training time, to circumvent the non-differentiability introduced by the intermediate sampling, the gradients for the generator were estimated using a REINFORCE-style procedure (Williams, 1992).

This intermediate hard selection facilitates the existence of tokens that do not have any contribution to the final prediction. While Lei et al. (2016) aimed for $\mathcal{S}_x$ to be the sufficient rationals for each prediction, the model might have learned an internal (emergent) communication protocol (Foerster et al., 2016)

IF *"very good"* IN *input*: SELECT *"very"* & RETURN 1;
IF *"not good"* IN *input*: SELECT *"not"* & RETURN 0.1;
IF *"good"* IN *input*: SELECT *"good"* & RETURN 0.8;
ELSE SELECT $\emptyset$ & RETURN 0.5.

Figure 2: Example of handshake.

that encodes information from the non-selected via the selected tokens, which we call a *handshake*. For example, the RCNN could learn a handshake such as the one in Figure 2, where the feature "good" was important in all three cases, but not selected in the first two.

**Eliminating handshakes.** Our goal is to gather a dataset $\mathcal{D}$ such that for all $x \in \mathcal{D}$, the set of non-selected tokens, which we denote $\mathcal{N}_x = x \setminus \mathcal{S}_x$, has zero contribution to the RCNN's prediction on $x$. Equivalently, we want to eliminate instances that contain handshakes. We show that:

$$\mathcal{S}_{\mathcal{S}_x} = \mathcal{S}_x \implies \text{no handshake in } x. \tag{3}$$

The proof is in Appendix B. On our example in Figure 2, on the instance "The movie was very good.", the model selects "very" and predicts a score of $1$. However, if we input the instance consisting of just "very", the model will not select anything[2] and would return a score of $0.5$. Thus, Equation 7 indeed captures the handshake in this example. From now on, we refer to non-selected tokens as irrelevant or zero-contribution interchangeably.

On the other hand, we note that $\mathcal{S}_{\mathcal{S}_x} \neq \mathcal{S}_x$ does not necessarily imply that there was a handshake. There might be tokens (e.g., *the* or *a* at the ends of the selection sequence(s)) that might have been selected in the original instance $x$ and that become non-selected in the instance formed by $\mathcal{S}_x$ without significantly changing the actual prediction. However, since it would be difficult to differentiate between such a case and an actual handshake, we simply prune the dataset by retaining only the instances for which $\mathcal{S}_{\mathcal{S}_x} = \mathcal{S}_x$.

**At least one clearly relevant feature.** With our pruning above, we ensured that the non-selected tokens have no contribution to the prediction. However, we are yet not sure that all the non-selected

---

[2] Our experiments show that the RCNN is capable of not selecting anything and providing its "bias" score as prediction. For example, this happened when we inputted sentences completely irrelevant to the task.

tokens are relevant to the prediction. In fact, it is possible that some tokens (such as "the" or "a") are actually noise, but have been selected only to ensure that the selection is a contiguous sequence (as we mentioned, the RCNN was penalized during training for selecting disconnected tokens). Since we do not want to penalize explainers for not differentiating between noise and zero-contribution features, we further prune the dataset such that there exists at least one selected token which is, without any doubt, *clearly relevant* for the prediction. To ensure that a given selected token $s$ is clearly relevant, we check that, when removing the token $s$, the absolute change in prediction with respect to the original prediction is higher than a significant threshold $\tau$. Precisely, if for the selected token $s \in \mathcal{S}_x$, we have that $|\text{encoder}(\mathcal{S}_x - s) - \text{encoder}(\mathcal{S}_x)| \geq \tau$, then the selected token $s$ is *clearly relevant* for the prediction.

Thus, we have further partitioned $\mathcal{S}_x$ into $\mathcal{S}_x = \mathcal{SR}_x \cup \mathcal{SDK}_x$, where $\mathcal{SR}_x$ are the clearly relevant tokens, and $\mathcal{SDK}_x$ are the rest of the selected tokens for which we do not know if they are relevant or noise (SDK stands for "selected don't know"). We see a diagram of this partition in Figure 3. We highlight that simply because a selected token alone did not make a change in prediction higher than a threshold does not mean that this token is not relevant, as it may be essential in combination with other tokens. Our procedure only ensures that the tokens that change the prediction by a given (high) threshold are indeed important and should therefore be ranked higher than any of the non-selected tokens, which have zero contribution. We thus further prune the dataset to retain only the datapoints $x$ for which $|\mathcal{SR}_x| \geq 1$, i.e., there is at least one clearly relevant token per instance.

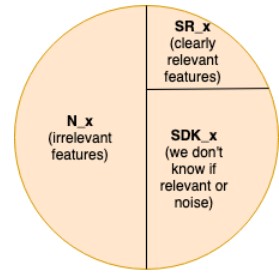

Figure 3: Partition of the features in our instances.

**Evaluation metrics.** First, we note that our procedure does not provide an explainer in itself, since we do not give an actual ranking, nor any contribution weights, and it is possible for some of the tokens in $\mathcal{SDK}_x$ to be even more important than tokens in $\mathcal{SR}_x$. However, we guarantee the following two properties:

**P1:** All tokens in $\mathcal{N}_x$ have to be ranked lower than any token in $\mathcal{SR}_x$.

**P2:** The first most important token has to be in $\mathcal{S}_x$.

We evaluate explainers that provide a ranking over the features. We denote by $r_1(x), r_2(x), \ldots, r_n(x)$ the ranking (in decreasing order of importance) given by an explainer on the $n = |x|$ features in the instance $x$. Under our two properties above, we define the following error metrics:

(A) **Percentage of instances for which the most important token provided by the explainer is among the non-selected tokens:**

$$\%\_first = \frac{1}{|\mathcal{D}_a|} \sum_{x \in \mathcal{D}_a} \mathbb{1}_{\{r_1(x) \in \mathcal{N}_x\}}, \tag{4}$$

where $\mathbb{1}$ is the indicator function.

(B) **Percentage of instances for which at least one non-selected token is ranked higher than a clearly relevant token:**

$$\%\_misrnk = \frac{1}{|\mathcal{D}_a|} \sum_{x \in \mathcal{D}_a} \mathbb{1}_{\{\exists i < j \text{ such that } r_i(x) \in \mathcal{N}_x \text{ and } r_j(x) \in \mathcal{SR}_x\}}. \tag{5}$$

(C) **Average number of non-selected tokens ranked higher than any clearly relevant token:**

$$avg\_misrnk = \frac{1}{|\mathcal{D}_a|} \sum_{x \in \mathcal{D}_a} \sum_{i < last\_si} \mathbb{1}_{\{r_i(x) \in \mathcal{N}_x\}}, \tag{6}$$

where $last\_si = \text{argmax}_j \{r_j(x) \in \mathcal{SR}_x\}$ is the lowest rank of the clearly relevant tokens.

Metric (A) shows the most dramatic failure: the percentage of times when the explainer tells us that the most relevant token is one of the zero contribution ones. Metric (B) shows the percentage of instances for which there is at least an error in the explanation. Finally, metric (C) quantifies the number of zero-contribution features that were ranked higher than any clearly relevant feature.

Table 1: Error rates of the explainers. The lower the values, the better the explainer. Best results in bold. In parenthesis are the standard deviations when averages are reported.

| Model | APPEARANCE | | | AROMA | | | PALATE | | |
|---|---|---|---|---|---|---|---|---|---|
| | %_first | %_misrnk | avg_misrnk | %_first | %_misrnk | avg_misrnk | %_first | %_misrnk | avg_misrnk |
| LIME | **4.24** | 24.39 | 7.02 (24.12) | 14.79 | 32.08 | 12.74 (33.54) | 2.92 | 13.93 | **3.48** (17.38) |
| SHAP | 4.74 | **16.81** | **1.16** (7.75) | **4.24** | **13.53** | **0.83** (7.10) | **2.65** | **9.20** | 9.25 (9.70) |
| L2X | 6.58 | 28.85 | 3.54 (12.66) | 12.95 | 31.61 | 4.41 (16.25) | 12.77 | 29.83 | 3.70 (13.05) |

## 5 RESULTS AND DISCUSSION

In this work, we instantiate our framework on the RCNN model trained on the BeerAdvocate corpus,[3] on which the RCNN was initially evaluated (Lei et al., 2016). BeerAdvocate consists of a total of $\approx .100K$ human-generated multi-aspect beer reviews, where the three considered aspects are appearance, aroma, and palate. The reviews are accompanied with fractional ratings originally between 0 and 5 for each aspect independently. The RCNN is a regression model with the goal to predict the rating, rescaled between 0 and 1 for simplicity. Three separate RCNNs are trained, one for each aspect independently, with the same default settings.[4]

With the above procedure, we gathered three datasets $\mathcal{D}_a$, one for each aspect $a$. For each dataset, we know that for each instance $x \in \mathcal{D}_a$, the set of non-selected tokens $\mathcal{N}_x$ has zero contribution to the prediction of the model. For obtaining the clearly relevant tokens, we chose a threshold of $\tau = 0.1$, since the scores are in $[0, 1]$, and the ground-truth ratings correspond to $\{0, 0.1, 0.2, \ldots, 1\}$. Therefore, a change in prediction of $0.1$ is to be considered clearly significant for this task.

We provide several statistics of our datasets in Appendix A. For example, we provide the average lengths of the reviews, of the selected tokens per review, of the clearly relevant tokens among the selected, and of the non-selected tokens. We note that we usually obtained 1 or 2 clearly relevant tokens per datapoints, showing that our threshold of 0.1 is likely very strict. However, we prefer to be more conservative in order to ensure high guarantees on our evaluation test. We also provide the percentages of datapoints eliminated in order to ensure the no-handshake condition (Equation 7).

**Evaluating explainers.** We test three popular explainers: LIME (Ribeiro et al., 2016a), SHAP (Lundberg & Lee, 2017), and L2X (Chen et al., 2018). We used the code of the explainers as provided in the original repositories,[5] with their default settings for text explanations, with the exception that, for L2X, we set the dimension of the word embeddings to 200 (the same as in the RCNN), and we also allowed training for a maximum of 30 epochs instead of 5.

As mentioned in Section 3, LIME and SHAP adhere to the feature-additivity perspective, hence our evaluation is not directly targeting these explainers. However, we see in Table 1 that, in practice, LIME and SHAP outperformed L2X on the majority of the metrics, even though L2X is a feature-selection explainer. We hypothesize that a major limitation of L2X is the requirement to know the number of important features per instance. Indeed, L2X learns a distribution over the set of features by maximizing the mutual information between subsets of $K$ features and the response variable, where $K$ is assumed to be known. In practice, one usually does not know how many features per instance a model relied on. To test L2X under real-world circumstances, we used as $K$ the average number of tokens highlighted by human annotators on the subset manually annotated by McAuley et al. (2012). We obtained an average $K$ of 23, 18, and 13 for the three aspects, respectively.

In Table 1, we see that, on metric (A), all explainers are prone to stating that the most relevant feature is a token with zero contribution, as much as $14.79\%$ of the time for LIME and $12.95\%$ of the time for L2X in the aroma aspect. We consider this the most dramatic form of failure. Metric (B) shows that both explainers can rank at least one zero-contribution token higher than a clearly relevant feature, i.e., there is at least one mistake in the predicted ranking. Finally, metric (C) shows

---

[3]http://people.csail.mit.edu/taolei/beer/

[4]https://github.com/taolei87/rcnn

[5]https://github.com/marcotcr/lime/tree/master/lime; https://github.com/slundberg/shap; https://github.com/Jianbo-Lab/L2X/tree/master/imdb-token.

that, in average, SHAP only places one zero-contribution token ahead of a clearly relevant token for the first two aspects and around 9 tokens for the third aspect, while L2X places around 3-4 zero-contribution tokens ahead of a clearly relevant one for all three aspects.

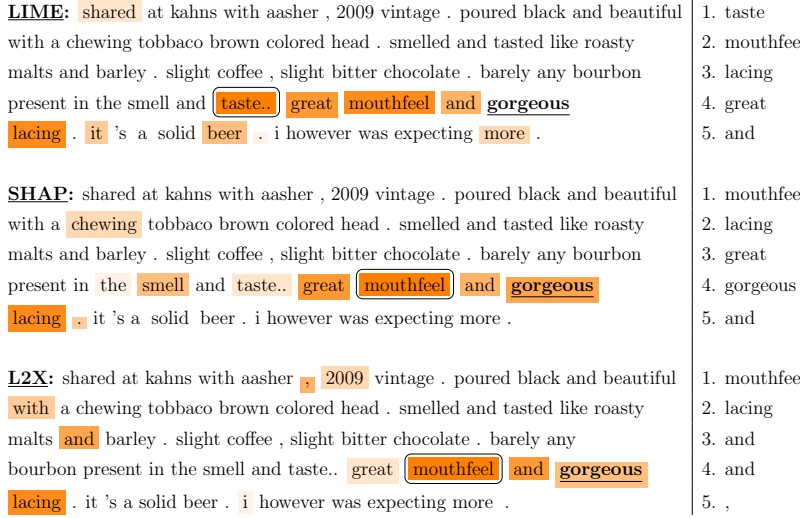

Figure 4: Explainers' rankings (with top 5 features on the right-hand side) on an instance from the palate aspect in our evaluation dataset.

**Qualitative Analysis.** In Figure 6, we present an example from our dataset of the palate aspect. More examples in Appendix C. The heatmap corresponds to the ranking determined by each explainer, and the intensity of the color decreases linearly with the ranking of the tokens.[6] We only show in the heatmap the first $K = 10$ ranked tokens, for visibility reasons. Tokens in $\mathcal{S}_x$ are in bold, and the clearly relevant tokens from $\mathcal{SR}_x$ are additionally underlined. The first selected by the explainer is marked wth a rectangular. Additionally the 5 ranks tokens by each explainer are on the right-hand side. Firstly, we notice that both explainers are prone to attributing importance to non-selected tokens, with LIME and SHAP even ranking the tokens "mouthfeel" and "lacing" belonging to $\mathcal{N}_x$ as first two (most important). Further, "gorgeous", the only relevant word used by the model, did not even make it in top 13 tokens for L2X. Instead, L2X gives "taste", "great", "mouthfeel" and "lacing" as most important tokens. We note that if the explainer was evaluated by humans assuming that the RCNN behaves reasonably, then this choice could have well been considered correct.

## 6 CONCLUSIONS AND FUTURE WORK

In this work, we first shed light on an important distinction between two widely used perspectives of explanations. Secondly, we introduced an off-the-shelf evaluation test for post-hoc explanatory methods under the feature-selection perspective. To our knowledge, this is the first automatic verification framework offering guarantees on the behaviour of a non-trivial real-world neural network. We presented the error rates on different metrics for three popular explanatory methods to raise awareness of the types of failures that these explainers can produce, such as incorrectly predicting even the most relevant token. While instantiated on a natural language processing task, our methodology is generic and can be adapted to other tasks and other areas. For example, in computer vision, one could train a neural network that first makes a hard selection of super-pixels to retain, and subsequently makes a prediction based on the image where the non-selected super-pixels have been blurred. The same procedure of checking for zero contribution of non-selected super-pixels would then apply. We also point out that the core algorithm in the majority of the current post-hoc explainers are also domain-agnostic. Therefore, we expect our evaluation to provide a representative view of the fundamental limitations of the explainers.

---

[6]While for LIME and SHAP we could have used the actual weights, for consistency, and since we evaluate the explainers only on their rankings, we keep the ranking-like heatmap.

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

## A  STATISTICS OF OUR GATHERED DATASETS

We provide the statistics of our dataset in Table 2. $N$ is the number of instances that we retain with our procedure, $|x|$ is the average length of the reviews, and $|\mathcal{S}_x|$, $|\mathcal{SR}_x|$, and $|\mathcal{N}_x|$ are the average numbers of selected tokens, selected tokens that give an absolute difference of prediction of at least $0.1$ when eliminated individually, and non-selected tokens, respectively. In parenthesis are the standard deviations. The column $\%(\mathcal{S}_{\mathcal{S}_x} \neq \mathcal{S}_x)$ provides the percentage of instances eliminated from the original BeerAdvocate dataset due to a *potential* handshake. Finally, $\%(|\mathcal{SR}_x| = 0)$ shows the percentage of datapoints (out of the non-handshake ones) further eliminated due to the absence of a selected token with absolute effect of at least $0.1$ on the prediction.

Table 2: Statistics of our datasets $\mathcal{D}_a$ for each aspect.

| Aspect | $N$ | $|x|$ | $|\mathcal{S}_x|$ | $|\mathcal{SR}_x|$ | $|\mathcal{N}_x|$ | $\%(\mathcal{S}_{\mathcal{S}_x} \neq \mathcal{S}_x)$ | $\%(|\mathcal{SR}_x| = 0)$ |
|---|---|---|---|---|---|---|---|
| APPEARANCE | 20508 | 145 (79) | 16.9 (8.4) | 1.33 (0.70) | 121 (56) | 15.9 | 73.2 |
| AROMA | 7621 | 139 (74) | 11.15 (6.48) | 1.16 (0.49) | 123 (57) | 72.0 | 58.0 |
| PALATE | 16494 | 153 (76) | 9.14 (5.38) | 1.21 (0.55) | 137 (59) | 39.2 | 66.5 |

## B  PROOF FOR NO HANDSHAKE CONDITION

We show that:
$$\mathcal{S}_{\mathcal{S}_x} = \mathcal{S}_x \implies \text{ no handshake in } x \,. \tag{7}$$

Proof: We note that if there is a handshake in the instance $x$, i.e., at least one non-selected token $x_k \in \mathcal{N}_x$ is actually influencing the final prediction via an internal encoding of its information into the selected tokens, then the model should have a different prediction when $x_k$ is eliminated from the instance, i.e., $\text{RCNN}(x) \neq \text{RCNN}(x - x_k)$. Equivalently, if $\text{RCNN}(x - x_k) = \text{RCNN}(x)$, then $x_k$ could not have been part of a handshake. Thus, if the RCNN gives the same prediction when eliminating *all* the non-selected tokens, it means that there was no handshake for the instance $x$, and hence the tokens in $\mathcal{N}_x$ have indeed zero contribution. Hence, we have that:

$$\text{RCNN}(x - \mathcal{N}_x) = \text{RCNN}(x) \implies \text{ no handshake in } x \,. \tag{8}$$

Since $x - \mathcal{N}_x = \mathcal{S}_x$, Equation 8 rewrites as:

$$\text{RCNN}(\mathcal{S}_x) = \text{RCNN}(x) \implies \text{ no handshake in } x \,. \tag{9}$$

From Equation 2, we further rewrite Equation 9 as:

$$\text{encoder}(\text{generator}(\mathcal{S}_x)) = \text{encoder}(\text{generator}(x)) \implies \text{ no handshake in } x \,. \tag{10}$$

Since, by definition, $\text{generator}(x) = \mathcal{S}_x$, we have that:

$$\text{encoder}(\mathcal{S}_{\mathcal{S}_x}) = \text{encoder}(\mathcal{S}_x) \implies \text{ no handshake in } x \,. \tag{11}$$

Hence, it is sufficient to have $\mathcal{S}_{\mathcal{S}_x} = \mathcal{S}_x$ in order to satisfy the right-hand-side condition of Equation 11, which finishes our proof. $\square$

## C    MORE EXAMPLES FROM OUR EVALUATION

**LIME:** nice brown " grolsch " like bottle ( good for re-use ) . **pours a dark yellow color with a lot of head** in the beginning which laces **well .** very fizzy **. smells** like fruit , maybe some apple and blueberry . no mouthfeel whatsover . besides being wet and a small initial alcohol , i could n't feel anything . tastes of fruit and not much alcohol , but i can start to feel a slight warming as i finish off the bottle . better than most american lagers , but very smooth . i think i would normally drink this too fast .

1. laces
2. yellow
3. lot
4. dark
5. of

**SHAP:** nice brown " grolsch " like bottle ( good for re-use ) . **pours a dark yellow color with a lot of head** in the beginning which laces **well .** very fizzy **. smells** like fruit , maybe some apple and blueberry . no mouthfeel whatsover . besides being wet and a small initial alcohol , i could n't feel anything . tastes of fruit and not much alcohol , but i can start to feel a slight warming as i finish off the bottle . better than most american lagers , but very smooth . i think i would normally drink this too fast .

1. fizzy
2. laces
3. head
4. nice
5. yellow

**L2X:** nice brown " grolsch " like bottle ( good for re-use ) . **pours a dark yellow color with a lot of head** in the beginning which laces **well .** very fizzy **. smells** like fruit , maybe some apple and blueberry . no mouthfeel whatsover . besides being wet and a small initial alcohol , i could n't feel anything . tastes of fruit and not much alcohol , but i can start to feel a slight warming as i finish off the bottle . better than most american lagers , but very smooth . i think i would normally drink this too fast .

1. laces
2. brown
3. lot
4. whatsover
5. nice

Figure 5: Explainers' rankings (with the top 5 features on the right-hand side) on an instance from the appearance aspect in our evaluation.

**LIME:** pours out very dark amber-red . almost no head , which dissappears quickly . " thick " aroma , rich maltiness , some toast , maybe a bit of cherry or some berry ( i do n't often eat berries of any type , makes it hard to descern ) . alcohol is pretty easy to pick up though . very rich malty flavour , a little sweet at first ; i get a hit of choclate malt , maybe very slightly fruity . alcohol quite prominent . sweet malt flavour falls off at finish and leaves mouth a little dry . mouthfeel quite thick , and more carbon dioxide in beer than is suggested by lack of head . all in all , i give it a pretty ok , its definately a strong flavour and aroma . got ta give it a little while for drinking , ca n't down this quick ( not that i 'd ever want to ) .

1. and
2. maltiness
3. ,
4. flavour
5. malt

**SHAP:** pours out very dark amber-red . almost no head , which dissappears quickly . " thick " aroma , rich maltiness , some toast , maybe a bit of cherry or some berry ( i do n't often eat berries of any type , makes it hard to descern ) . alcohol is pretty easy to pick up though . very rich malty flavour , a little sweet at first ; i get a hit of choclate malt , maybe very slightly fruity . alcohol quite prominent . sweet malt flavour falls off at finish and leaves mouth a little dry . mouthfeel quite thick , and more carbon dioxide in beer than is suggested by lack of head . all in all , i give it a pretty ok , its definately a strong flavour and aroma . got ta give it a little while for drinking , ca n't down this quick ( not that i 'd ever want to ) .

1. aroma
2. aroma
3. rich
4. some
5. ,

**L2X:** pours out very dark amber-red . almost no head , which dissappears quickly . " thick " aroma , rich maltiness , some toast , maybe a bit of cherry or some berry ( i do n't often eat berries of any type , makes it hard to descern ) . alcohol is pretty easy to pick up though . very rich malty flavour , a little sweet at first ; i get a hit of choclate malt , maybe very slightly fruity . alcohol quite prominent . sweet malt flavour falls off at finish and leaves mouth a little dry . mouthfeel quite thick , and more carbon dioxide in beer than is suggested by lack of head . all in all , i give it a pretty ok , its definately a strong flavour and aroma . got ta give it a little while for drinking , ca n't down this quick ( not that i 'd ever want to ) .

1. aroma
2. rich
3. little
4. which
5. .

Figure 6: Explainers' rankings (with the top 5 features on the right-hand side) on an instance from the aroma aspect in our evaluation.

