# OpenReview forum: "Can I Trust the Explainer? Verifying Post-Hoc Explanatory Methods"
_ICLR.cc/2020/Conference — Reject_

### Official Review · AnonReviewer3 · 2019-10-23
**Official Blind Review #3**

**Rating:** 3

**Review:**

Summary -

The paper proposes a verification method for instance wise feature explanations. The verification framework uses an RCNN to identify two types of tokens a) the tokens that are not predictive of outcome b) the subset of clearly relevant tokens for prediction. The data used for RCNN is a pruned version of the data used to train the  black-box. The pruning eliminates data points to ensure that the tokens not selected by RCNN have no contribution to the outcome and that the model does not exhibit suffer from learning "handshakes". A handshake is defined as the set of tokens that may be spuriously missed because their information is encoded in another relevant token. This proof to identify such data points is shown and the RNN is therefore expected to be able to reliably identify 3 kinds of tokens a) Those that have zero contribution to the outcome. b) Those that definitely have some contribution to the outcome and c) those that could be relevant or noisy. Three instance-wise feature selection methods are compared. Results are provided on 3 metrics. a) % instances for which the most important tokes provided by the explainer is among the non-selected tokens, b) % of instances for which at least one non-selected token is ranked higher than a relevant token, and c) Average number of non-selected tokens ranked higher than any obviously relevant tokens.

Clarifications and concerns:
1. For the dataset considered here, I would like to see the distribution of the irrelevant, clearly relevant and unsure if they are relevant tokens as detected by the RCNN. How does this change if I further prune the dataset after ensuring that handshake and other issues have been eliminated. The main concern I have is the idea of verifying other explanations using a neural network itself. I can train the RCNN neural network with half the data (and satisfy the properties the authors mention) and my evaluation would change significantly. From the appendix I see that most of the tokens could be in the set $SDR_x$.

2. What if the set of tokens don't overlap between the RCNN and the black-box to be verified. That said, I think the assumptions of the framework should be much more explicitly mentioned.

3. The std deviations in the experiments are very high. Can the authors justify this and how it is still okay to use this framework for evaluating feature importance based explanations.

Minor:
1. You have cited the "Anchors" paper twice?
2. Page 3 - typo - "....explainer should provide different explanations for the trained model on real data than when the data..."
-----------------------------------------------------------------------------------------------------------------------------------------------------------------------
Update -

I have read the authors response.
If the pruned dataset is created using an RCNN, then it is not clear if the RCNN is used to just explain itself or all other methods as well. Like I said, if we just train the model on a slightly different distribution of labels, or half the data randomly sampled irrespective of labels, the explanations will change because the pruned dataset i.e. the ground truth may significantly change. I am still not convinced how this makes for a good verification framework to asses other explainers.

It is also unclear how generalizable this verification process is to other domains. I will therefore not be updating my score.

**Experience Assessment:**

I have read many papers in this area.

**Review Assessment: Checking Correctness Of Derivations And Theory:**

I assessed the sensibility of the derivations and theory.

**Review Assessment: Checking Correctness Of Experiments:**

I carefully checked the experiments.

**Review Assessment: Thoroughness In Paper Reading:**

I read the paper at least twice and used my best judgement in assessing the paper.

---

> ### Author Response · Authors · 2019-11-07
> **RE Official Blind Review #3**
>
>
> A1. Our general answer should help in clarifying this. The RCNN itself doesn’t detect 3 types of tokens, but only 2: selected and non-selected. It is our procedure that provides the pruned datasets for which (1) the non-selected are guaranteed to be irrelevant, and (2) we further identify clearly relevant features among the selected ones.
> We train the RCNN only once (per aspect) on the original datasets. The pruning procedure is applied after the RCNN was trained in order to obtain an associated pruned dataset (for each trained RCNN) on which we provide the above 2 guarantees. If one wants to train another RCNN (on half of the original data, or on any other subset, or even if one simply changes the seed), then one has to do the pruning procedure again to obtain a new pruned dataset associated to the newly trained model. This new pair of (trained model, pruned dataset) would constitute another instance of evaluation test for the explainers, with potentially different results. But this would not invalidate the results that we obtained on the 3 trained models we evaluated on.
>
>
> A2. Usually, “black-box” would refer to the model to be explained by the explainers, while the method “to be verified”, in this work, would be the explainer. So, it is not clear what “black-box to be verified” refers to. In case this refers to the model to be explained, then the RCNN is not used to explain any other model than itself, as we highlighted in our general answer. If “black-box to be verified” refers to the explainer to be verified, then this is precisely our goal, to penalize the explainer based on the difference between the features that it considered relevant and the ground-truth relevant ones.
>
> A3. The stdev reported in Table 1 for the avg_misrnk metric is simply showing the amount of variability in the error that the explainers make on this metric. The fact that the explainers are making a highly variable amount of errors shows a downside of these explanatory methods which is emphasized by our framework, rather than a downside in our framework. Since we are not introducing an explainer but an evaluation, the high variability is not a concern for our framework.

---

### Official Review · AnonReviewer1 · 2019-10-24
**Official Blind Review #1**

**Rating:** 3

**Review:**

-------------------- AFTER
The original rating of "Weak Reject" still holds as the authors failed to provide proper justification for the raise concerns and support their claims through additional experiments.

"We do not introduce an explanation generation framework, as explainers do. " - The proposed evaluation requires the explainer of the NLP model to agree with the RCNN in-terms of relevant or irrelevant words, to be considered a good explainer. The RCNN model which is defining the relevant and irrelevant tokens for a prediction task is in fact stating that we can explain the decision of an NLP model in terms of relevant and irrelevant tokens. Hence, the proposed RCNN can also be considered as an explainer. The evaluation task is demonstrating if the other explainers are providing explanations consistent with this new explainer based on RCNN.


"The RCNN is not meant to explain any other models except itself." - Unclear

"Regarding the request for more experiments:" - The authors don't provide enough justification to "why they didn't perform more experiments?"

" Hence, with our current instantiations, any domain-agnostic explainer can be evaluated" - The experiment to validate this claim are missing.

"The novelty of our paper consists in the fact that, to our knowledge, it is the first to (1) shed light over a fundamental difference" - This is not a technical novelty. This is an exploratory analysis based observation

"and (2) propose a methodology for evaluating explainers that ...and without human intervention (unlike evaluation type 4)." -  In Section 5 Qualitative Analysis, the authors are also doing human evaluation like other methods in evaluation type-4 of their related works. Also, doing human evaluation is a strong way to justify an explainer. Though expensive, whenever possible it should be done and is in no way a limitation of current evaluation metrics.

Your model needs labelled data for training RCNN. This adds a constraint on the usability and scalability of your proposed evaluation method. Since RCNN is also black-box, one will required another explainer to explain the RCNN.

In the worst-case scenario, if RCNN is trained with data such that it considers all relevant words as irrelevant, the evaluation made by RCNN will be incorrect. Hence, " Success depends on the ability of the RCNN to extract correct subsets of tokens."



-------------------  BEFORE
The paper proposed a verification framework to evaluate the performance of different explanatory methods in interpreting a given target model. Specifically, the authors evaluated three explanatory methods namely, LIME, SHAP and L2X for a target model trained to perform sentiment analysis on text data. Authors assume for each input text, there is a subset of tokens that are most relevant and that are completely irrelevant to the final prediction task.  The proposed framework uses a recurrent convolutional neural network (RCNN) to find these subsets. The performance of an explainer is evaluated in terms of overlap between the RCNN most relevant tokens and the most relevant tokens provided by the explainer as an explanation.

Major
•	The paper lack technical novelty.
•	The proposed architecture uses a RCNN to find the most relevant subset of tokens. Firstly, RCNN is also a black box that provides no intuition behind its selection decision. Secondly, in the absence of the ground truth labels for true relevance and irrelevance of a token in input sentence, this explainer method can also suffer from “assuming a reasonable behavior” assumption. The method assumes that the RCNN is performing reasonably in identifying relevant subsets.
•	The success of the method depends on the ability of the RCNN to extract correct subsets of tokens. The data used for training the RCNN, might have some underlying bias. In that case, the evaluation is not accurate.
•	In related work, for “Interpretable target models” the authors mentioned LIME as an example of explainer functions that explains target models that are “very simple models may not be representative for the large and intricate neural networks used in practice”. LIME locally explains the decision of a complex function for a given data point using simpler models like linear regression. But LIME itself can be used for generating explanation for prediction of complex neural network like Inception Net.
•	The example used to explain the difference between feature additive and feature selection-based explainer methods, is confusing. Its not clear how in health diagnostics, one will prefer feature-selection perspective. Although the most relevant features used for the instance are important to understand the decision, but in clinical settings sometimes low rank features can also be useful to understand the target model.
•	For text, the relevant features are the individual tokens of the input sentence. Similarly, for images relevance can be important regions of the image. The authors did not have any experiments on images or tabular data.
•	In the experiment section, the comparison is made with only 3 explainer models and for just one task. The experiments are inadequate.
•	In Figure 4, the colormap is not readable.


**Experience Assessment:**

I have read many papers in this area.

**Review Assessment: Checking Correctness Of Derivations And Theory:**

N/A

**Review Assessment: Checking Correctness Of Experiments:**

I assessed the sensibility of the experiments.

**Review Assessment: Thoroughness In Paper Reading:**

I read the paper at least twice and used my best judgement in assessing the paper.

---

> ### Author Response · Authors · 2019-11-07
> **RE Official Blind Review #1**
>
> REVIEWER: Lacks technical novelty.
> ANSWER: The novelty of our paper consists in the fact that, to our knowledge, it is the first to (1) shed light over a fundamental difference in the goals of two major types of explanations that are currently being directly compared despite their distinct goals, and (2) propose a methodology for evaluating explainers that  does not make speculations on the behaviour of the model (unlike evaluation type 3, see our related work - Section 2), which is based on a real-world scenario (complex neural architectures and real-world datasets) (unlike evaluation types 1 and 2), and without human intervention (unlike evaluation type 4). Consequently, we are also the first to test current state-of-the-art explainers in a setting having all the above features, pointing out some critical explainers' deficiencies.
>
>
> R: Lack of intuition behind RCNN’s selection decision and assuming reasonable behavior of the RCNN.
> A: The general answer should help in clarifying this. The lack of intuition behind the RCNN’s selection decision does not affect the fact that, in the end, the non-selected tokens were irrelevant to the model’s predictions on the pruned datasets, nor that the clearly relevant tokens we identify are indeed among the ground truth relevant tokens for the trained RCNNs.
> We also do not assume that the RCNN has reasonable behavior in identifying correct subsets. The two pruning mechanisms in Section 4 ("eliminating handshakes" and "at least one clearly relevant feature") are precisely how we guarantee (as opposed to just assume) that for each pair of (trained model, pruned evaluation dataset) the non-selected tokens are indeed irrelevant in the prediction of the trained models on the instances in their associated pruned datasets, and that there is a subset of at least 1 clearly relevant tokens in each instance of the pruned datasets. The behavior might also not be reasonable, as we explain in the answer below.
>
>
> R: Success depends on the ability of the RCNN to extract correct subsets of tokens.
> A: We do not assume any correctness (wrt the original task) of the selected tokens. Our evaluation is accurate even if the RCNN learns an underlying bias. Our handshake check ensures that the instances where a potential bias influences the prediction are either eliminated or the bias is reflected in the selected tokens. For example, suppose that the RCNN learned the bias: [if ‘bottle’ in input_text, then predict 1] (‘bottle’ is a neutral word that should not indicate the most positive sentiment score of 1). If for an instance containing ‘bottle’, the RCNN doesn’t select ‘bottle’ (which was its decision criteria), the instance would be detected as a handshake and eliminated from its associated pruned dataset. If the RCNN selects ‘bottle’, there are 2 possibilities: (1) we fail to identify it as clearly relevant, in which case our evaluation wouldn’t penalize the explainer for flagging it as relevant; or (2) we identify ‘bottle’ as clearly relevant, in which case our evaluation would penalize the explainer only if it ranks any irrelevant tokens higher than ‘bottle’ (since that shouldn’t be the case). In either of the cases, the evaluation is faithful to the trained model’s behavior, regardless of what its behavior is.
>
>
> R: Authors mentioned LIME as an example of explainer that explains “very simple models may not be representative for the large and intricate neural networks used in practice”.
> A: In the referred paragraph, we wrote that LIME was evaluated based on how correct it explains a simple model. Of course, LIME can be used to explain any complex model, and this is why we actually test it on a complex model like the RCNN.
>
>
> R: The example in Fig 1 is confusing. It’s not clear how in health diagnostics, one will prefer feature-selection perspective.
> A: Our example is only meant to show the difference between the two perspectives of explanations, and not that one perspective is better than the other. We now deleted the sentence where we expressed our own opinion that one might prefer the feature selection in individual decision-making tasks, as it was indeed simply our opinion and does not affect the rest of the work.
>
>
> R: More experiments
> A: Please see our general answer.
>
>
> R: Colormap is not readable.
> A: We will update it to make it more readable.

---

> > ### Comment · AnonReviewer1 · 2019-11-16
> > **REPLY**
> >
> > The original rating of "Weak Reject" still holds as the authors failed to provide proper justification for the raise concerns and support their claims through additional experiments.
> >
> > "We do not introduce an explanation generation framework, as explainers do. " - The proposed evaluation requires the explainer of the NLP model to agree with the RCNN in-terms of relevant or irrelevant words, to be considered a good explainer. The RCNN model which is defining the relevant and irrelevant tokens for a prediction task is in fact stating that we can explain the decision of an NLP model in terms of relevant and irrelevant tokens. Hence, the proposed RCNN can also be considered as an explainer. The evaluation task is demonstrating if the other explainers are providing explanations consistent with this new explainer based on RCNN.
> >
> >
> > "The RCNN is not meant to explain any other models except itself." - Unclear
> >
> > "Regarding the request for more experiments:" - The authors don't provide enough justification to "why they didn't perform more experiments?"
> >
> > " Hence, with our current instantiations, any domain-agnostic explainer can be evaluated" - The experiment to validate this claim are missing.
> >
> > "The novelty of our paper consists in the fact that, to our knowledge, it is the first to (1) shed light over a fundamental difference" - This is not a technical novelty. This is an exploratory analysis based observation
> >
> > "and (2) propose a methodology for evaluating explainers that ...and without human intervention (unlike evaluation type 4)." -  In Section 5 Qualitative Analysis, the authors are also doing human evaluation like other methods in evaluation type-4 of their related works. Also, doing human evaluation is a strong way to justify an explainer. Though expensive, whenever possible it should be done and is in no way a limitation of current evaluation metrics.
> >
> > Your model needs labelled data for training RCNN. This adds a constraint on the usability and scalability of your proposed evaluation method. Since RCNN is also black-box, one will required another explainer to explain the RCNN.
> >
> > In the worst-case scenario, if RCNN is trained with data such that it considers all relevant words as irrelevant, the evaluation made by RCNN will be incorrect. Hence, " Success depends on the ability of the RCNN to extract correct subsets of tokens."

---

### Official Review · AnonReviewer2 · 2019-10-28
**Official Blind Review #2**

**Rating:** 3

**Review:**

Overview/Contribution:
====================
The authors present a explanation generation framework that help validate post-hoc explanations when the explanations are generated based on feature selection. They claim to demonstrate their method by showing failure modes of exiting explanation generation methods.

Overall, the paper is not ready to be accepted to the conference and I describe my rational with the following strengths and weaknesses.

Strength:
========
+ Explanations make models more transparent and easy to understand for end users of the decision made by complex models such as deep neural networks [1]. In that respect, having a verification mechanism for post-hoc explanations is interesting and useful.
+ The paper is easy to read and follow.
Weakness:
===========
- evaluating explanations generated for an opaque model with another opaque model (RCNN) is cyclical.
- Just like many literature in this nascent space, interpretation (which is measuring the contribution of features or subsets of features towards predicted output) is confused as explanation. Human level explanations don’t necessarily depend on the direct interaction or contribution of model derived features. Rather they describe ‘why’ the model come up with the decision produced.
- Explanation generation is gaining traction in the deep learning community especially for critical applications such as healthcare and security. However, the authors claim that post-hoc explanations currently are only evaluated for only simple non-neural model. That is misleading given the recent attention toward generating explanations for various deep learning models.
- As a generalized pos-hoc explanation generators verification framework, the experiments are seriously lacking and are not well designed to illicit broad applicability.

1) Bekele, E., Lawson, W. E., Horne, Z., & Khemlani, S. (2018). Implementing a Robust Explanatory Bias in a Person Re-identification Network. In Proceedings of the IEEE Conference on Computer Vision and Pattern Recognition Workshops (pp. 2165-2172).

**Experience Assessment:**

I have published one or two papers in this area.

**Review Assessment: Checking Correctness Of Derivations And Theory:**

I assessed the sensibility of the derivations and theory.

**Review Assessment: Checking Correctness Of Experiments:**

I carefully checked the experiments.

**Review Assessment: Thoroughness In Paper Reading:**

I read the paper at least twice and used my best judgement in assessing the paper.

---

> ### Author Response · Authors · 2019-11-07
> **RE Official Blind Review #2**
>
>
> REVIEWER: Evaluating explanations generated for an opaque model with another opaque model (RCNN) is cyclical.
> ANSWER: Our general answer should help clarify this. The RCNN is not meant to explain other opaque models, it only explains itself, hence there is no cycle. We only evaluate explainers on the trained RCNNs with their associated pruned datasets for which we provided the guarantees mentioned in the general answer. The RCNN has a degree of transparency that we exploit: it itself, selects the features that it will further exclusively use in the final prediction. Our 2 pruning procedures ensure that, on the instances of the pruned datasets, the RCNN’s selection faithfully represents the model’s inner-working: the non-selected tokens are indeed irrelevant and some of the selected tokens are clearly relevant.
>
> R: Authors claim that post-hoc explanations currently are only evaluated for simple non-neural model.
> A: We do not claim that explainers are only evaluated on non-neural models. In the related work, we had listed the 4 types of evaluations we identified in the literature, the last 3 of which are based on complex neural models, but they have other downsides. We have updated the paper to ensure that we mention all 4 types of evaluations together in all parts of the paper.
>
> R: More experiments
> A: Please see general answer.
>
> R: Referenced human-level explanation paper
> A: Thank you for mentioning it, we added it accordingly in the related work.

---

### Author Response · Authors · 2019-11-07
**General Answer**



We thank the reviewers for their insightful comments. It seems that most of the raised concerns are misunderstandings that can be resolved with the following clarification.

We do not introduce an explanation generation framework, as explainers do. Instead, we introduce a methodology for generating evaluation tests for those explainers. Our tests consist of pairs of (trained model, pruned evaluation dataset) with 2 guarantees on the behaviour of each trained model over the instances in its associated pruned dataset:
the non-selected tokens are irrelevant for each prediction,
for each instance in the pruned dataset, we identify one subset of clearly relevant tokens.
Based on these guarantees, we evaluate explainers only on these pairs of (trained model, pruned evaluation dataset). The models are trained only once on the whole original dataset, while each pruned dataset is dependant on its associated trained model and is used only for evaluating the explainers. The RCNN is the architecture of our trained models. The RCNN is not meant to explain any other models except itself.

Regarding the request for more experiments: First, our methodology is domain-agnostic, so we open the path for the community to instantiate it in any area and generate many more evaluation tests. We gave 3 instantiations on an NLP task and our experiments proved that well-known explainers can make critical errors. For example, they can even tell us that the most important feature is one that was totally irrelevant, which is particularly problematic in safety-critical applications. Secondly, most of the explainers in the literature are also domain-agnostic. Hence, with our current instantiations, any domain-agnostic explainer can be evaluated by applying it to the 3 pairs of (trained model, pruned dataset) that we will release.

---

### Decision · Program_Chairs · 2019-12-19

**Decision:**

Reject

**Comment:**

The paper proposes a framework for generating evaluation tests for feature-based explainers. The framework provides guarantees on the behaviors of each trained model in that non-selected tokens are irrelevant for each prediction,  and for each instance in the pruned dataset, one subset of clearly relevant tokens is selected.

After reading the paper, I think there are a few issues with the current version of the paper:

(1) the writing can be significantly improved: the motivation is unclear, which makes it difficult for readers to fully appreciate the work. It seems that each part of the paper is written by different persons, so the transition between different parts seems abrupt and the consistency of the texts is poor. For example, the framework is targeted at NLP applications, but in the introduction the texts are more focused on general purpose explainers. The transition from the RCNN approach to the proposed framework is not well thought-out, which makes the readers confused about what exactly is the proposed framework and what is the novelty.

(2) the claimed properties of the proposed framework are rather straightforward derivations. The technical novelty is not as high as claimed in the paper.

(3) The experiment results are not fully convincing.

All the reviewers have read the authors' feedback and responded. It is agreed that the current version of the paper is not ready for publication.